# Nano-Enabled Antivirals for Overcoming Antibody Escaped Mutations Based SARS-CoV-2 Waves

**DOI:** 10.3390/ijms241713130

**Published:** 2023-08-23

**Authors:** Aminur Rahman, Kumar Jyotirmoy Roy, Gautam Kumar Deb, Taehyeong Ha, Saifur Rahman, Mst. Khudishta Aktar, Md. Isahak Ali, Md. Abdul Kafi, Jeong-Woo Choi

**Affiliations:** 1Department of Microbiology and Hygiene, Bangladesh Agricultural University, Mymensingh 2202, Bangladesh; aminur50651@bau.edu.bd (A.R.); jkroy39833@bau.edu.bd (K.J.R.); saifurrahman@bau.edu.bd (S.R.); khudishta.27952@bau.edu.bd (M.K.A.); isahakalimb@gmail.com (M.I.A.); 2Department of Biotechnology, Bangladesh Livestock Research Institute, Dhaka 1341, Bangladesh; debgk2003@yahoo.com; 3Department of Chemical and Biomolecular Engineering, Sogang University, 35 Baekbeom-ro, Mapo-gu, Seoul 04107, Republic of Korea; hatae91@sogang.ac.kr

**Keywords:** SARS-CoV-2, mutations, variants, clinical resurgence, nanotechnology

## Abstract

This review discusses receptor-binding domain (RBD) mutations related to the emergence of various SARS-CoV-2 variants, which have been highlighted as a major cause of repetitive clinical waves of COVID-19. Our perusal of the literature reveals that most variants were able to escape neutralizing antibodies developed after immunization or natural exposure, pointing to the need for a sustainable technological solution to overcome this crisis. This review, therefore, focuses on nanotechnology and the development of antiviral nanomaterials with physical antagonistic features of viral replication checkpoints as such a solution. Our detailed discussion of SARS-CoV-2 replication and pathogenesis highlights four distinct checkpoints, the S protein (ACE2 receptor coupling), the RBD motif (ACE2 receptor coupling), ACE2 coupling, and the S protein cleavage site, as targets for the development of nano-enabled solutions that, for example, prevent viral attachment and fusion with the host cell by either blocking viral RBD/spike proteins or cellular ACE2 receptors. As proof of this concept, we highlight applications of several nanomaterials, such as metal and metal oxide nanoparticles, carbon-based nanoparticles, carbon nanotubes, fullerene, carbon dots, quantum dots, polymeric nanoparticles, lipid-based, polymer-based, lipid–polymer hybrid-based, surface-modified nanoparticles that have already been employed to control viral infections. These nanoparticles were developed to inhibit receptor-mediated host–virus attachments and cell fusion, the uncoating of the virus, viral gene expression, protein synthesis, the assembly of progeny viral particles, and the release of the virion. Moreover, nanomaterials have been used as antiviral drug carriers and vaccines, and nano-enabled sensors have already been shown to enable fast, sensitive, and label-free real-time diagnosis of viral infections. Nano-biosensors could, therefore, also be useful in the remote testing and tracking of patients, while nanocarriers probed with target tissue could facilitate the targeted delivery of antiviral drugs to infected cells, tissues, organs, or systems while avoiding unwanted exposure of non-target tissues. Antiviral nanoparticles can also be applied to sanitizers, clothing, facemasks, and other personal protective equipment to minimize horizontal spread. We believe that the nanotechnology-enabled solutions described in this review will enable us to control repeated SAR-CoV-2 waves caused by antibody escape mutations.

## 1. Introduction

The emergence of SARS-CoV-2 variants with every clinical wave of the COVID-19 pandemic has appeared as a global challenge. Mutations of various amino acid residues at the receptor motif of the spike (S) protein are considered to be the major cause of the emergence of several variants [1]. Based on many shared attributes and mutation characteristics of the genome, the WHO has classified all the SARS-CoV-2 variants as a (i) variant of concern (VOC), (ii) variant of interest (VOI), or (iii) variant under monitoring (VUM) [2,3]. Depending on the transmission rate, Alpha, Beta, Gamma, Delta, Epsilon, and Omicron were termed as “Variants of concern” (VOCs) [4]. Now, the Omicron variant is further classified into five major lineages such as BA.1, BA.2, BA.3, BA.4, and BA.5. Among those lineages, BA.2, BA.4, and BA.5 have also been declared as VOCs according to ECDC 2023 [2,5]. In addition, considering disease severity, vaccine neutralization ability, and receptor-binding domain (RBD) mutation tendency, Epsilon, Eta, Iota, Kappa, and Zeta are declared as VOI [5]. Several recent variants of Omicron, such as BA.2.75 (x), BQ.1, XBB (z), and XBB.1.5-like(a), have been categorized as VOI [2,3,4], while other lineages like CH.1.1, XBB.1.16, and XBB.1.5-like + F456L are categorized as VUM. However, the CDC categorized all the variants as VUM except Omicron [2,4,6]. Most regions have, therefore, already gone through two or three phases of outbreaks, which come in repetitive waves with short pauses in between. Mutations of the viral genome, which allow the virus to escape neutralizing antibodies, have been suggested to be the major cause of such repetitive outbreaks [7], and the receptor-binding domain (RBD) of the viral S protein has been reported to be the primary site for such mutations, usually appearing following an outbreak or immunization [8]. The RBD region is the major motif responsible for establishing host cell–virus interactions that initiate viral replication [9]. This important domain has already undergone several mutations, resulting in the repeated waves of clinical outbreaks the world has seen [7], which is why most of the developed vaccine candidates are unable to ensure solid protection. It is well understood that vaccinated populations produce both neutralizing and non-neutralizing antibodies, with the neutralizing antibody providing immunity against the infection [10]. However, the mutated virus escapes immunity by re-adjusting its attachment motif and replication pathways [8]. Such readjustments through mutations in the RBD sequence help the virus evade neutralizing antibodies. Antibody escape mutations are, therefore, considered the most vital mechanisms behind the repeated emergence of clinical waves of SARS-CoV-2. For example, the Pfizer/BioNTech (BNT162b2) and Moderna vaccines conferred 96% protection against the original Wuhan virus but only 86.3% protection against the Alpha variant [11]. Likewise, the BNT162b2 vaccine exhibited 75.0%, 50.34%, and 40% protection against the Beta, Gamma, and Delta variants, respectively [7,12,13]. Despite the ability of new variants to escape neutralizing antibodies, the currently available vaccines significantly reduced mortality rates in clinically affected patient groups [13], with studies reporting reductions in fatality of about 80% in vaccinated compared to non-vaccinated populations [14]. The presence of non-neutralizing antibodies in vaccinated populations may be responsible for the reduction in fatalities, through the inhibition of the interstitial spread of the virus. Vaccination should therefore be continued, even though none of the currently available vaccines offer solid protection. Researchers across the globe have thus been focusing on advanced technological solutions that can target specific checkpoints in the intracellular replication and extracellular spreading processes of coronaviruses. Very recently, nanotechnology approaches have successfully been used for the development and preparation of the BNT162b2, mRNA-1273, NVX-CoV2373, EpiVacCorona, Vaxfectin^®^, Cervarix^®^, Inflexal^®^V, Epaxal^®^, and Dermavir vaccines against SARS-CoV-2 [15,16].

However, more advanced, sustainable technological solutions are needed to control the still ongoing pandemic. Therefore, in this review, we focus on applications of nanotechnological principles as a sustainable means for tackling repeated waves of COVID-19 through the development of antiviral nanostructures with physical antagonistic features against the RBD. Such nanostructures could physically block the RBD of spike proteins, and blocked RBDs would not be able to interact with ACE2 receptors during the host cell attachment. Mutations in amino acid residues of the RBD and other amino acids, such as the D614 G sequence [17], determine differences in the spread of different corona variants [8,18]. For example, the Alpha (B.1.1.7) variant spreads at a 43–82% higher rate than SARS-CoV-2 [12], while the spread of the South African Beta variant (also termed B.1.351) surpassed that of the Alpha variant by 50% [19]. Likewise, the Gamma variant spread at a 50% higher rate than the Beta variant [20], while the transmission rate of the Delta variant in India was twice that of the Gamma variant [21]. Transmission rates can be reduced through a quickly applied test–track–treat (TTT) strategy of carriers with the help of high-performance sensing devices such as wearable sensor devices, epidermal electronics, and implantable sensors [21]. The World Health Organization (WHO) has already introduced a number of rapid diagnostic tools [22] developed for this purpose, such as antigen-detecting rapid diagnostic tests (Ag-RDTs), nucleic acid amplification tests (NAAT), and lateral flow immunoassays (LFI) [23]. However, none of these methods can be applied to the fast-tracking of huge populations, which is especially essential in developing countries because they are expensive, time-consuming, and not very sensitive. Faster, more sensitive, and cost-effective diagnostic methods are therefore needed. High-functional sensing devices based on nanotechnological principles, such as nano-biosensors, bioelectronics, and nano-biochips, are ideal candidates for this purpose, as they allow for real-time, highly sensitive monitoring of patients from a distance.

In this review, we thoroughly discuss viral structures associated with cell fusion and the replication process to highlight the points where therapeutic or preventive approaches could be applied. Our review also focuses on mutations of various spike proteins and the emergence of variants with such escape mutations. Several nanomaterial-based inhibitions of viral checkpoints are discussed. Finally, we identify four specific checkpoints: the S protein (ACE2 receptor coupling), the RBD motif (ACE2 receptor coupling), ACE2 coupling, and the S protein cleavage site for the development of antiviral nanomaterials, nanocarriers, and nano-biosensors that could help tackle repeated COVID-19 waves.

## 2. Viral Structures Associated with Cell Fusion

SARS-CoV-2 is an enveloped, spherical or pleomorphic, non-segmented, positive-sense, single-stranded RNA virus; it varies in size, ranging from 80 to 220 nm in diameter (Figure 1a) [24]. The viral genome codes for several structural (Figure 1b) and non-structural proteins (Figure 1b) [25], among them glycoprotein structures, namely spike (S) protein, and non-glycoprotein structures, such as membrane protein (M), which reside in the virus.

The M glycoprotein plays a key role in transmembrane budding, whereas heavily glycosylated S glycoprotein is used as a ligand for membrane fusion in the initiation of viral entry. The S protein plays a vital role in the receptor recognition mechanisms underlying the membrane fusion process, as illustrated in Figure 2A. Among the two subunits of the S protein, the S1 subunit possesses an RBD [26] that recognizes host receptor ACE2 and other NTD for glycan and other co-receptor bindings [27], while the S2 protein (HR1 and HR2) stabilizes the RBD domain (Figure 2B) [28]. Unfortunately, the RBD of S1 protein, an important receptor motif, mutates frequently, thereby resulting in the emergence of variants, as illustrated in Figure 1c. However, considering its unique roles in the viral fusion process, we thoroughly discuss the structural and functional aspects of the S protein to highlight the checkpoints against which antiviral nanomaterials could be designed. Other structural proteins, such as nucleocapsid (N) and envelope protein (E), are also discussed to uncover their specific roles in virus replication. The N protein is one of the most abundant viral proteins expressed in the host at an early stage of infection, is involved in viral RNA genome organization for progeny viruses, and has hydrophobic features that are essential for viral assembly [29].

Genome diversity analysis of SARS-CoV-2 has revealed similarities with other human coronavirus strains, such as SARS, SARS2, SAR_S_, MERS, HKU1/OC43, HCoV-229E, HCoV-NL63, and HCoV-HKU1 [30]. While the distribution of structural and non-structural proteins in its genome is similar to that observed in SARS-CoV and MERS-CoV, it is the furin-like cleavage site of its S protein that is responsible for the extreme spread [31]. Distinct variations in the furin-like cleavage site have been highlighted as a target for therapeutic strategies. One more difference between the amino acid sequences of SARS-CoV, MERS-CoV, and SARS-CoV-2 has been identified in the receptor-binding motif (RBM) that is involved in the ACE2 receptor-activated viral adhesion process, as depicted in Figure 2A [25,32]. It has been reported that residues of the RBM, namely Ans501 and Gln493, are interacting with human ACE2, suggesting that the capacity for human-to-human transmission of SARS-CoV-2 resides in them [25]. The unique claw-like structure on the outer surface of the RBM of SARS-CoV-2 has also been found to be involved in virus-ACE2 coupling during cell virus fusion, which makes it a potential target for nanotherapeutic approaches, as illustrated in Figure 2B [33]. Furthermore, specific amino acids at positions 442, 472, 479, 480, and 487 enhance viral binding with human ACE2, and other amino acids in these regions have been found to enhance adhesion to palm civet ACE2 [29,31,34]. These findings suggest that nanotechnology could be employed to develop nanoparticles functionalized with amino acids that inhibit cell–receptor bonding and thereby prevent the viral adhesion process.

## 3. Escape Mutations and Clinical Waves of COVID-19

Detailed investigations of different COVID-19 waves in their respective geographic locations have revealed close associations with mutations of virion at its spike glycoprotein (see the preceding sections as well as Figure 1 and Table 1). Such mutations have led to the emergence of several variants, such as B.1.1.7, B.1.351, P.1, B.1.617, B.1.1.529, BA.1, BA.2, BA.3, BA.4, BA.5, BA.2.75 (x), BQ.1, XBB (z), XBB.1.5-like, CH.1.1, XBB.1.16, and XBB.1.5-like + F456L [5,8,35,36]. Most of these variants appeared to be more infectious and virulent than the original Wuhan virus.

It has been reported that antibodies developed from natural infection with a variant or through vaccination are less effective at neutralizing other mutants [8,16,39,40]. The emergence of such variants has been attributed to several mutations in the SARS-CoV-2 spike protein, i.e., in the K417, L452, K477, T478, E484, Q498, and N501 region of the RBD, as depicted in Table 1 [9,35,41]. On the basis of single-nucleotide polymorphisms (SNPs), alterations in amino acid residues in the same RBD region are considered a major cause of the mutation-dependent emergence of variants [41]. In the case of Alpha-N501Y, a residue N amino acid at position 501 was replaced with a Y, whereas Beta-K417N, -E484K, and -N501Y originated from mutations at residues 417, 484, and 501, respectively, where K, E, and N were replaced with N, K, and Y acid [14]. Likewise, Gamma-K417T, -E484K, and -N501Y exhibited mutations at residues 417, 484, and 501, where K, E, and N acids were replaced with T, K, and Y acids, respectively [40]. In the Delta-K417N, -L452R, -T478K, and -E484Q strains, mutations occurred at positions 417, 452, 478, and 484, with N, R, K, and Q replacing K, L, T, and E, respectively [13,35,41,42]. Similarly, in Omicron-K417N, -K477N, -T478K, -E484A, -Q498R, and -N501Y, K, T, E, Q, and N at residues 417, 477, 478, 484, 498, and 501 were replaced with N, K, A, R, and Y [35,43]. All mutants exhibit the ability to escape neutralizing antibodies, and the vaccines developed as of today, therefore, do not offer full immune protection [8]. More immune-escape mutations might emerge while the pandemic situation progresses [8], and while antibody responses to SARS-CoV-2 receptor-binding sites are strong enough to neutralize the original Wuhan strain, mutated variants can elude this response. Hence, it does not seem feasible to prevent the spread of SARS-CoV-2 through regular updates of the available vaccines in response to the emergence of new mutants, which is why the whole world is looking for alternative technological solutions to the ongoing pandemic. In this review, we suggest focusing on recently popularized, powerful nanotechnology applications to develop a sustainable strategy to diagnose, treat, and immunize against COVID-19.

## 4. Importance of Nanotechnology

Nanotechnology is the method for controlling molecules below 100 nm scale for enhancing desired functionality. In the material world, nanomaterial lies in the scale of ≥1 nm to ≤100 nm. In the 21st century, nanotechnology has been considered the most attractive tool in many fields, including engineering, biology, chemistry, and physics [44]. Nowadays, nanomaterial science, electronics and nanoscale engineering (ENE), nano-agriculture, nanomedicine, nano-biotechnology (NBT), nano-robotics, nano-machines, and nano-toxicology have been established as branches of nanotechnology [44,45,46,47,48,49,50]. Nanotechnology offers numerous advantageous features for many biomedical applications, like enhanced functionality through their increased volume aspect ratio and durability in action and targeted delivery through precise selectivity [51,52,53] (Figure 3).

Recently, nanotechnology has emerged as one of the most promising technologies on account of its ability to deal with viral diseases in an effective manner, addressing the limitations of traditional antiviral medicines. It has not only enabled us to overcome problems related to the solubility and toxicity of drugs but also imparted unique properties to drugs, which in turn has increased their potency and selectivity toward viral particles against the host cells [54]. Overall, antivirals coated with nanoparticles offer several advantageous features compared to non-coated antivirals, like increased cellular uptake capability of drugs due to increased ion exchangeability of NPs, decreased doses of drugs due to precise selectivity of NPs through targeted delivery of drugs, increased cellular influx and decreased efflux, increased durability of action of Nanoparticle coated drugs through their slow release, and increased antiviral activity through targeted modification of functional groups [44,46,53].

In spite of its many advantages, this exciting technology still has many limitations, such as the unavailability of biocompatible, biodegradable, and eco-friendly nanomaterials [55,56,57], and most chemically synthesized metal and metal oxide nanomaterials being unsuitable for application in biological systems [56]. Their stability and durability of action is another challenge, because of their relatively short half-life [55]. The biocompatibility and toxicity of inorganic nanoparticles should thus be assessed before applications are implemented in living systems. Biocompatibility assays could be performed using in vitro cell culture systems or in vivo live animal models. In vitro virus neutralization tests are essential to confirm antiviral activity, and in vivo live animal models are needed to determine physical, biological, and histopathological changes as well as the efficacy, safety, half-life, and shelf-life of nano-drugs. Eco-friendly green synthesis protocols using naturally available materials could be an alternative to chemical synthesis processes. The desired shelf-life of a nanomedicine could be adjusted by controlling the size, shape, charge, and surface chemistry of the nanomaterials. Likewise, toxicity could be minimized by adjusting the particle size: for example, 1.4 nm-sized Au NPs and Ag NPs are toxic, while 15 nm-sized NPs are nontoxic for living systems [58]. Lack of knowledge and awareness about the use of nanomedicine is another limitation of this promising technology. Therefore, this review calls for future research to mitigate the challenges discussed above for the safe application of this technology in impeding viral pandemics, with a special focus on COVID-19 resurgences.

## 5. Antiviral Nanomaterials That Prevent Viral Infections

Most nanoparticles exert antiviral effects through the inhibition of (i) receptor-mediated host–virus attachments and cell fusion [59], (ii) the uncoating of the virus [59,60], (iii) viral gene expression [61], (iv) protein synthesis [62,63], (v) assembly of progeny viral particles [64,65], and (vi) release of virion [66,67]. Various target-specific inhibitory roles of different nanoparticles are described in the following.

### 5.1. Inhibition of Receptor-Mediated Host–virus Attachments and Cell Fusion

Receptor-mediated virus attachment, cell fusion, and entry are the initial steps for viral replication. The virus possesses many glycoprotein receptors anchored with capsid that are projected through the envelope to establish communication between host cell receptors during the attachment process [35]. In the case of the SARS-CoV-2, the S1 subunit of the S protein of the spike glycoprotein receptor attaches with the ACE2 of the host cell to initiate the attachment process [68]. So, the viral S1—ACE2 blocking cloud is an important checkpoint for inhibiting virus entry into the host cell. Therefore, this review emphasized nanoparticle-assisted ACE2 receptor and viral S protein blocking through the development of nanoparticles with physical antagonistic features against the ACE2 receptor or S1 protein. Towards this direction, researchers across the globe have targeted the cell–virus adhesion step as an important checkpoint for the inhibition of viral replication and pathogenesis [69]. Several nanoparticles, including Ag NPs, Au NPs, ZnO NPs, CuO NPs, graphene oxide nanoparticles (GO NPs), carbon dots (CDs), as well as lipid and carbohydrate nanoparticles and their nanohybrids, have been suggested and tried out for this inhibition process, as shown in Figure 4. Ag NPs have been found to inhibit the attachment of human immunodeficiency virus-1 (HIV-1) and respiratory syncytial virus (RSV) envelope proteins to the host cell by blocking glycoprotein [69,70]. Several other studies have also reported that some modified Ag NPs with mercaptoethanol sulfonate, tannic acid, and antiviral oseltamivir inhibit the attachment of herpes simplex virus-1 (HSV-1), HSV-2, and influenza virus (H1N1) to the host cell [70,71]. Likewise, Au NPs have been shown to inhibit the attachment of HIV, HSV-1, and H1N1 to lymphocytes, macrophages, and endothelial cells in the brains of mice [72,73,74]. Many oxide nanoparticles, such as CuO NPs, ZnO NPs, and polyglycerol sulfate-coated GO NPs, inhibit the attachment and entry of the hepatitis C virus (HCV), HSV-2, and African swine flu virus into host cells [75,76,77], while carbon nanostructures, like fullerene and CDs, functionalized with boronic acid and 4-aminophenyl boronic acid, inhibit entry of HSV-1, HCoV-229E, H1N1, and porcine epidemic diarrhea virus into the host cell [78,79,80]. Researchers have also applied peptides, polypeptides, and antiviral functionalized nanoparticles, such as β-CD-PACM nanoparticles loaded with acyclovir, hydrophilic N-(2-hydroxypropyl)-3-trimethylammonium chitosan chloride (HTCC), hydrophobically modified HTCC (HM-HTCC), N-[1-(2,3-Dioleoyloxy)propyl]-N,N,N-trimethylammonium (DOTAP) liposomes, phosphatidylserine (PS) liposomes, phosphatidylcholine (PC) liposomes, polyanionic carbosilane dendrimers, and stearylamine-coated liposome nanoparticles to various viral adhesion processes, and found that these nanohybrids inhibit the attachment of HIV, HSV-1, HSV-2, human coronaviruses HCoV-NL63, and murine hepatitis virus (MHV) to the host cell [81,82,83].

### 5.2. Inhibition of Virus Uncoating

Immediately after internalization, the virion coat dissolves due to cytosolic enzyme reactions, resulting in the exposure of the viral genome, which is known as the uncoating stage of viral replication [84]. Uncoating is considered the second most important checkpoint for inhibiting viral replication [84,85]. Therefore, viral encapsulation with nanomaterials could protect the capsid from degradation with cytosolic enzyme activity. Thus, biocompatible viral encapsulating material could be an ideal solution to prevent the multiplication of the virus. Focusing on this, Iron oxide nanoparticles (IONPs), GO NPs, solid-lipid nanoparticles (SLNs), and nano-capsules have been employed to target this checkpoint, inhibit the uncoating process, and thereby impede virus replication [70,86,87]. Figure 5 illustrates these processes. It has been reported that Ag NPs inhibit the uncoating of the Tacaribe virus by blocking its receptor glycoprotein structures [59,60,87,88]. Likewise, carbon-based nanoparticles, such as GO NPs and carbon nanotubes (CNT), have been employed against H1N1 to inhibit uncoating through physical encapsulation of the viral glycoprotein coat, as shown in Figure 5a [89]. Antiviral-coated lipid nanoparticles, for example, nano-capsules entrapped with azidothymidine-triphosphate (AZT-TP), polymers coated with polyethyleneimine, β-cyclodextrin-poly (4-acryloylmorpholine) mono-conjugate (β-CD-PACM), and SLNs loaded with atazanavir, have also been found to inhibit HIV uncoating through targeted delivery of the trapped drug into the cytoplasm [85]. A few recent studies have reported that IO NPs and GO NPs inhibit the uncoating of SARS-CoV-2 through irreversible changes to S1-RBD induced by the formation of IO NP-S1-RBD complexes [86].

### 5.3. Inhibition of Viral Gene Expression

Following the uncoating stage, the viral genome replicates to form numerous copies that are later translated into structural and non-structural viral proteins [35,90]. After entry and uncoating, the virus releases its RNA genome in the cytoplasm for gene expression [91]. In this process, positive-sense ssRNA was translated to form ORF-1a and ORF-1b [92]. The ORF was polymerized through RdRp to form dsRNA, and thus, the gene expression occurred. So, inhibition of RdRp-mediated polymerization could be a way to impede gene expression. Thus, this step has also been considered as a checkpoint for inhibiting viral replication. Therefore, nanoparticle-assisted blocking of the RdRp enzyme could also be an ideal choice to prevent viral replication. Several nanoparticles, including CuO NPs, chitosan nanoparticles (Chi NPs), Au NPs, ZnO NPs, GO NPs, Se NPs, CNTs, and CDs have been employed to inhibit viral gene expression at this checkpoint [93,94,95,96], as shown in Figure 6. Studies have reported that CuO NPs inhibit genome expression of Emiliania huxleyi virus 86, HSV, poliovirus, and influenza A through reactive oxygen species (ROS)-mediated oxidative damage of the viral genome [96,97]. Copper nanohybrid particles, such as copper–iodide, gold–copper, and copper–nanostructures, have been used to inhibit HuCoV-229, H1N1, and SARS-CoV viral gene expression; they do so by damaging mRNA and inactivating proteases and polymerase enzymes [94,98]. Likewise, ZnO NPs and polyethylene glycol (PEG)-coated ZnO NPs inhibit H1N1, nidoviruses, and the RNA expression of other viral genes through ROS-mediated inactivation of RNA-dependent RNA polymerase [94,99]. Additionally, Se NPs loaded with siRNA and Chi NPs coated with siRNA inhibit Ebola virus 71 and influenza viral gene expression through targeted delivery of siRNA to the VP1 gene [54,100]. Different carbons and their nanohybrids, such as GO NPs, CDs, CNTs, fullerene derivatives, and quanta dots (QDs), have also been found to inhibit HCV, SARS-CoV, RSV, the pseudorabies virus, the porcine epidemic diarrhea virus, HIV, influenza viruses, and other RNA viruses. Carbon nanohybrids inhibit viral genome replication through activation of IFN-α, alterations of viral proteins, generation of ROS, and inactivation of protease and reverse transcriptase enzymes [98,101]. Furthermore, many antivirus-coated lipid nanoparticles, including PEG-PLGA loaded with V-ATPase, liposomes coated with ivermectin, and solid lipid nanoparticles loaded with adefovir, have been tested at the uncoating checkpoint and found to inhibit H1N1, H3N2 [102], dengue virus, West Nile virus, yellow fever virus [103], and hepatitis B virus (HBV) [104] viral gene expression.

### 5.4. Inhibition of Protein Synthesis

During gene expression, positive-sense RNA transcript serves as a template for viral protein [32]. The translation of viral transcript is, therefore, considered another vital checkpoint for virus replication [12,32]. Therefore, nanoparticle-assisted inhibition of protein synthesis through blocking ribosomal RNA would also be the ideal choice for preventing virus replication. Bearing this in mind, many researchers used Ag NPs, IO NPs, and Se NPs to inactivate protease and polymerase enzyme-mediated translation processes [12], as illustrated in Figure 7(bi). Furthermore, Ag NPs and polysaccharide-coated Ag NPs have been reported to inhibit glycoprotein synthesis of the Tacaribe and monkeypox viruses [105,106]. Several oxide nanoparticles, including ZnO NPs and IO NPs, have shown similar inhibitory effects on the protein synthesis of H1N1 influenza and HCV by inactivating peroxidase and catalase enzyme activities [107,108], while Se NPs loaded with antiviral drugs also inhibit the protein synthesis of H1N1 [52,85]. Several carbon-based nanoparticles, such as CNTs functionalized with protoporphyrin IX (PPIX) and fullerene derivatives, likewise inhibit the protein synthesis of influenza viruses through RNA degradation and that of HIV by interacting with Vpr, Nef, and Gag proteins [109,110,111]. Additionally, antivirus-coated polymeric nanoparticles such as amantadine-loaded micelles and siRNA-coated PLGA nanoparticles hinder the protein synthesis of H1N1 and HSV-2 by blocking hemagglutinin protein and complementary mRNA strands [112].

### 5.5. Inhibition of the Viral Particle Assembly

The assembly of viral particles involves the transportation of chemically distinct macromolecules through different pathways, such as through interactions between proteins of viral and cellular origin, between viral proteins and nucleic acids, and among viral proteins themselves, to complete the virion. Thus, the assembly of newly synthesized viral protein into the rough endoplasmic reticulum could be another checkpoint for impeding viral replication. Focusing on this, many researchers employed Au NPs, IO NPs, and GO to inhibit such processes at this key checkpoint [113,114,115,116], as illustrated in Figure 7(bii). It has been reported that Au NPs inhibit the assembly of influenza virus, HIV, and HSV proteins, as well as those of other viral particles, by blocking their interactions [117,118]. Likewise, IO NPs coated with poly hexamethylene biguanide inhibit the assembly of HSV-1, viral hemorrhagic septicemia virus, and infectious pancreatic necrosis virus through irreversible damage of viral particles [115], while GO NPs inhibit the porcine epidemic diarrhea and pseudorabies viruses’ protein assemblies through physical interactions between graphene derivatives and virus particles [113,116].

### 5.6. Inhibition of Virion Release

The last step of the replication pathways is the release of progeny virions, either by lysis of the host cell or through extrusion processes [22]. This step is also considered an important checkpoint for nanoparticle-mediated inhibition of replication to tackle the spread of infectious viruses [119]. Ag NPs, GO NPs, and dendrimers have been applied to inhibit this step by blocking various viral structural proteins [67,120], as shown in Figure 7(biii). Ag NPs inhibit the release of infant HCV extracellular virion through interactions with structural proteins [121]. Likewise, GO NPs inhibit the release of HSV-1 through physical interactions with enveloped proteins [122], and erythrocyte membrane-coated spiky nanostructures impede the release of the influenza A virus by blocking the outer shell of the infant virus [64]. Additionally, dendrimers coated with pentaerythritol derivatives inhibit the release of HIV and enterovirus 71 (EV71) virion by blocking vesicular membranes [123].

## 6. Use of Nanomaterials in Targeted Drug Delivery

Nanoparticle-mediated targeted drug delivery is another emerging tool for antiviral therapies. Several organic and inorganic nanoparticles have been utilized to deliver drugs to target checkpoints for selective actions, such as the inhibition of viral infections by avoiding unwanted exposure to other cellular and subcellular organelles [51] (see Figure 8a and Table 2). A Vero cell-based in vitro study revealed that Chi NPs coated with siRNA (Chi-siRNA NPs) inhibit influenza virus replication and protect 50% of mice against a lethal challenge through targeted delivery of siRNA [124]. Another study reported that siRNA NPs released siRNA into the primary site of infection, which minimizes systemic siRNA loss and avoids toxicity while protecting mice against lethal influenza, HSV, cancer cells, and SARS-CoV-2 challenges [125,126,127].

## 7. Use of Nanomaterials in Vaccine Preparations

Vaccination is the most reliable way to prevent and eradicate deadly infectious diseases. Several of the effective vaccines against COVID-19, including those developed by Pfizer, Moderna Vaxfectin^®^, Cervarix^®^, Inflexal^®^V, Epaxal^®^, Dermavir, and Novavax, employ nanotechnology principles to selectively target specific actions and minimize adverse reactions [90,142,143]. These engineered nanovaccines enhance the immunization potential of the bioactive peptide, increase antibody titers, improve the T- and B-cell immune response, and increase the stability and half-life of the vaccine. These enhancements are achieved because of their unique properties, such as hydrophobicity, increased surface areas, ion exchange ability, capacity to cross biological barriers, and ability to inhibit viral protein synthesis and replication [144]. The mRNA-based vaccines developed by BioNTech/Pfizer and Moderna employ positively charged lipid nanoparticles as vaccine carriers, exhibit increased stability, and are resistant to RNase-mediated degradation and boosting of both humoral and cellular immune responses via inducing the lymphatic system against SARS-CoV-2 infection [21]. A number of nanoparticle-based vaccines such as Vaxfectin^®^, Cervarix^®^, Inflexal^®^V, Epaxal^®^, and Dermavir have recently been developed against potentially deadly viruses such as H5N1, HIV, HAV, HBV, and HPV [85]. It has been reported that chitosan-loaded nanovaccines reduce lung virus titers and nasal viral shedding of H1N1 and induce cross-reactivity of mucosal IgA and cellular immune responses in the respiratory tract, resulting in a 100% reduction in morbidity [145,146]. Likewise, spike nanovaccines conjugated with adjuvants increase immune responses against MERS-CoV and SARS-CoV-1 via targeted delivery of proteins to T- and B-cells [57,147]. A novel nanovaccine called Self-Assembling Protein-based Nanoparticles (SANPs), conjugated with monomeric proteins, decreases RSV load in the lungs via the activation of T-cells in mouse models [148,149]. The virus-like nano-capsule embedded with viral capsid has been tested against HBV core and bacterial capsids to induce a defensive mechanism that increases cytotoxic responses of T-cells without side effects [150,151]. The modified nanovaccine conjugated with a palivizumab-targeted epitope (called FsII) reduces RSV load while enhancing immune responses through targeted delivery to N proteins [152,153]. Additionally, a novel nanovaccine conjugated with PLGA and DEPE-PEG polymers increases prophylactic action against MERS-Cov through targeted delivery of a subunit of viral antigen to the infected cell [154,155,156].

## 8. Scope of Nanotechnology in Controlling Clinical Waves of COVID-19

### 8.1. Development of Nano-Biosensors

Nano-biosensors are considered an attractive tool worldwide, because they enable the fast and sensitive real-time monitoring of analytes, incorporating biomedical devices that have already been used in the remote monitoring of biophysical parameters such as pulse rates, heart rates, oxygen levels, and pH levels. For example, BIOTEST AG, single-walled carbon nanotubes (SWCNTs), surface plasmon resonance (SPR), plasmonic photothermal (PPT) biosensors, localized surface plasmon resonance (LSPR), surface-enhanced Raman scattering (SERS), and fluorescence-based nano-biosensors have been developed for the detection of HIV, HPV, H1N1, dengue virus, SARS-CoV-2, and other viruses [157].

Wearable devices consisting of multisensory electrodes, including pressure, heat, oxygen, pulse, respiratory, and PH sensors, can measure body parameters such as pulse rates, pressure, or temperatures (Figure 9). Healthcare devices such as sensor patches (e.g., a band-aid adhesive patch used for glucose monitoring [158]), epidermal electronics (e.g., used to detect electrophysiological signals on the epidermis [128,159]), and contact lenses with embedded electronics (such as sensors, transmitters, and amplifiers used for health monitoring [160]) have recently attracted attention because of their potential to be applied in biomedical settings. Additionally, skin-equivalent sensors (such as the SkinEthic^TM^, Lyon, Franch, MatTek, Ashland, MA 01721, USA, StrataTech, St. Louis, MO, USA) [127,161] or bio-implantable sensors (such as specific absorption rate (SAR)), implantable blood pressure sensors, medical implant communication service (MICS), etc. [162,163]) combined with distance-monitoring devices could be useful for the diagnosis of COVID-19. However, applications of such devices are challenging due to inadequate interactions at the skin–device interface that lead to poor signal acquisition, and they are hence unfit for distance monitoring. Additionally, state-of-the-art devices often exhibit bio-incompatibility issues resulting from adverse tissue reactions, such as erythema, itching, and inflammation, which can cause severe discomfort for users. The signal acquisition efficacy of sensing devices is another challenging aspect that could, however, be tackled through in vitro and in vivo experimentation. The shelf life of such sensors also needs to be determined before they can be applied in real-life settings. Engineering solutions with biocompatible skin equivalent (SE)-embedded multi-electrode sensors that can establish biological communication between the skin and wearables and achieves the signal sensitivity necessary for monitoring clinical parameters of COVID-19 patients from a safe distance are therefore needed. Nano-biosensor-based wearable sensors, skin-equivalent electronics, epidermal electronics, and implantable sensors might be candidates for such distance monitoring devices. Nano-biosensor-based self-monitoring devices could serve as a TTT tool to identify symptomless carries among large populations and reduce horizontal virus transmission. Physicians and other healthcare personnel using such monitoring devices will be able to monitor patients with COVID-19 from a distance without being exposed.

### 8.2. Development of SARS-CoV-2-Neutralizing Nanoparticles

Nanoparticles have been studied in many fields of biomedical sciences for their increased surface area, excellent sensitivity, and enhanced functionality [164,165,166]. They have been suggested as an alternative to antibiotics [167], antifungals [168], and antivirals [169,170] to curb the use of these drugs. The ion exchange ability, enhanced functionality, ion absorption capability, and chemical complexation of multifunctional nanoparticles promise to be effective in neutralizing SARS-CoV-2. However, many nanoparticles exhibit compromised bio-compatibility because of the materials used, underlying chemical synthesis processes, and improper functionalization of target ligands. Nanoparticles from biocompatible, biodegradable, and eco-friendly materials synthesized through green processes could be effective antivirals for controlling the SARS-CoV-2 pandemic.

Nanostructures with RBD-like physical antagonistic features could serve as effective therapeutic agents that block the RBD from the inhibition of ACE2 receptor-mediated cell fusion. NPs functionalized antagonistic nanostructure encapsulated RBD formed that will block the specific RBD site, resulting in the inhibition of RBD-ACE2 adhesion as shown in Figure 10. Mutation-dependent alterations of amino acid residues can then not impact the inhibition process, which may prevent a rapid spread as well as the pathogenicity of variants caused by mutations of the S protein. Considering these differences to previous approaches, nanomaterial-based therapeutics using nanoscale hybrid structures with ACE2 receptor-like antagonism features on their surfaces that can neutralize SARS-CoV-2 well ahead of its adhesion to the ACE2 receptor could serve as an alternative treatment for clinical cases (see Figure 11).

More specifically, nanoparticles functionalized with anti-salicylic acid will be effective in neutralizing viruses circulating in a multicellular host and thus prevent further disease progression (Figure 11). After entering the circulation, anti-salicylic acid-functionalized nanoparticles will be coupled with viruses to inhibit their attachment to the ACE2 receptor on the cell surface, which will then inhibit the fusion process. This means that the ACE2-activated angiotensin regulation mechanism will remain uninterrupted, and homeostasis will be maintained in the cardiovascular system. Nano-biosensor-based early detection of infections will also enable nanoparticle-assisted neutralization of SARS-CoV-2 during primary viremia and thus prevent fatalities.

### 8.3. Development of Nanoscale Antiviral Drugs and Vaccine Carriers

Many antiviral organic and inorganic nanoparticles, as well as their composite derivatives, have already been used as antiviral agents. Several antiviral drugs functionalized with nanoparticles, such as nano-capsules embedded with lipid nanoparticles and protein nanoparticles prepared from polymers, dendrimers, or micelles, have been applied as antivirals as well as vaccine carriers against cytomegalovirus, HIV, the Ebola, Zika, and dengue viruses, coronaviruses, HBV, and HCV [99,114,171,172,173]. A number of drug nanocarriers have been introduced for other purposes, for example, lipid-based nanocarriers for targeted therapies [174], RNA and protein nanocarriers for cancer and cellular niche therapy [54]. In the same way, tissue-specific drug carriers can be developed for the delivery of drugs against COVID-19 (Figure 12). Patients experiencing respiratory distress can be treated with a nanocarrier probed for lung tissue, while one probed for renal tissue or the cardiovascular system can be employed for patients experiencing renal or cardiovascular dysfunction. Such targeted delivery avoids not only unwanted exposure of unaffected systems to antivirals but also side effects.

Although several countries have developed vaccines against SARS-CoV-2, the duration of the protection they offer is under debate. Even natural antibodies of a recovered patient do not protect from subsequent re-infection. Synthetic nanoparticle-functionalized antibodies can therefore be a choice for neutralizing SARS-CoV-2. Neutralizing antibodies against viral S protein-coated nanoparticles can be developed to avoid ACE2-mediated cell fusion [175]. Likewise, antibodies developed against RBD can also be used for nanoparticle functionalization to interfere with viral replication [54,175]. Overall, the immune response against SARS-CoV-2 antigen can be enhanced with immune-targeted nanotherapeutics, such as biocompatible polymeric, lipid-based, or inorganic NPs, because of their ion exchange capacities [176] and their ability to pass through all sorts of barriers (e.g., the blood–brain, placental, and articular capsule barriers [90]). Nanoparticle-assisted immune enhancements have been reported for graphene [90], nanodiamonds [176], carbon nanotubes [177], polystyrene particles [178], and other nanoparticles. On the other hand, some nanomaterials (e.g., GO and alum) exert immunomodulatory effects on innate immune mechanisms [179,180].

## 9. Summary and Conclusions

This review focuses on nanomaterial-assisted therapeutical ways to tackle repetitive waves of COVID-19. We discuss detailed nanostructural aspects of SARS-CoV-2 in relation to the virus pathogenesis and clinical manifestations and highlight specific checkpoints for inhibiting viral replication, intervening in the disease progression, and slowing the spread of the infection. Our detailed discussion of state-of-the-art diagnostics and therapeutics reveals the potential improvements that could be achieved by employing functional nanomaterials. Considering the specificity and enhanced functionality of nanotechnological products, nanotherapeutic agents can be developed for neutralizing viruses both in the host and in the environment. This review, therefore, emphasizes three specific scopes for employing nanotherapeutics to interfere with viral replication: the development of (i) anti-spike protein nanoparticles (NPs), (ii) anti-furin nanoparticles, and (iii) anti-RBD nanoparticles. Anti-spike protein nanoparticles can be developed through the functionalization of anti-sialic acid to prevent the fusion of the virus with the host cell. Anti-furin nanoparticles can be developed using S protein to inhibit furin-like cleavage to minimize the transmission of SARS-CoV-2. Anti-RBD nanoparticles can also be developed using specific amino acids of the RBD to inhibit cell–receptor bonding and prevent viral adsorption. As proof of this concept, we discuss the antiviral actions of several nanoparticles against many potentially deadly viruses through the inhibition of host–virus attachment, uncoating, gene expression, protein synthesis, assembly, and release of the virion. For example, Ag NPs, Chi NPs, Au NPs, ZnO NPs, CuO NPs, GO NPs, IO NPs, CDs, lipid and carbohydrate nanoparticles, SLNs, nano-capsules, Se NPs, carbon nanotubes, polymeric nanoparticles, fullerene nanostructures, as well as dendrimers and their nanohybrids have been employed to inhibit the replication of HIV, HSV, HBV, H1N1, SERS-CoV, MERS-CoV, and other potentially deadly viruses. Moreover, many nanomaterials (lipids and proteins, dendrimers, micelles, polymers) have been employed to minimize the adverse effects of antiviral drugs by reducing doses through targeted delivery. Virus-neutralizing NPs with receptor-like antagonistic surface features can be developed to neutralize SARS-CoV-2 both in the host (minimizing clinical features) as well as in the environment (reducing the virus spread). Although state-of-the-art diagnostics can confirm SARS-CoV-2 infections, they cannot be used to screen large numbers of patients in developing countries because of their time requirements and high costs. The development of self-healthcare devices that allow for fast and sensitive real-time monitoring is therefore critical. For such purposes, cost-effective and sensitive self-health monitoring sensor patches, skin equivalent/wearable/implantable/epidermal electronic/sensor-embedded contact lenses, and similar devices can be developed for TTT mass applications, particularly to prevent the spread of SARS-CoV-2. We, therefore, suggest focusing on research programs that are necessary to develop quick and sensitive nano-diagnostics and high-functional nanotherapeutics that will enable us to tackle the still ongoing COVID-19 pandemic.

## Figures and Tables

**Figure 1 ijms-24-13130-f001:**
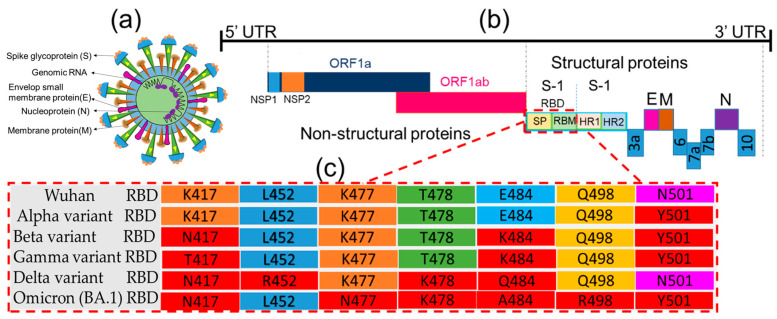
Distribution of structural and non-structural proteins in the SARS-CoV-2 genome where (**a**) Structural and associated proteins, (**b**) gemon organization of structural and non-structural proteins, and (**c**) Mutation site of the gens of structural protein.

**Figure 2 ijms-24-13130-f002:**
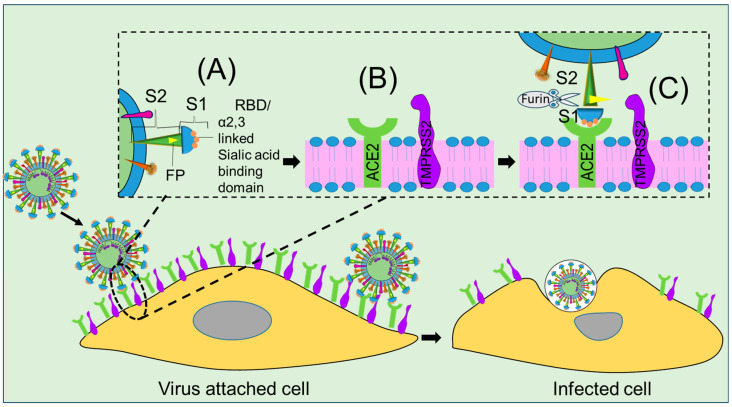
Nanostructural view of SARS-CoV-2 spike glycoprotein (S) and the potential mechanism underlying host cell fusion. (**A**) initiation of host-virus attachments involving S1,S2, FP of S protein and sialic acid binding domain, (**B**) ACE2 receptor and TMPRSS2 of hot cell, and (**C**) cell fusion through cleavage of furin protein.

**Figure 3 ijms-24-13130-f003:**
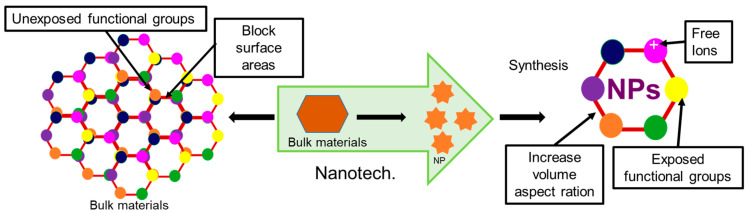
Advantages of the nanoparticle compared to their bulk material.

**Figure 4 ijms-24-13130-f004:**
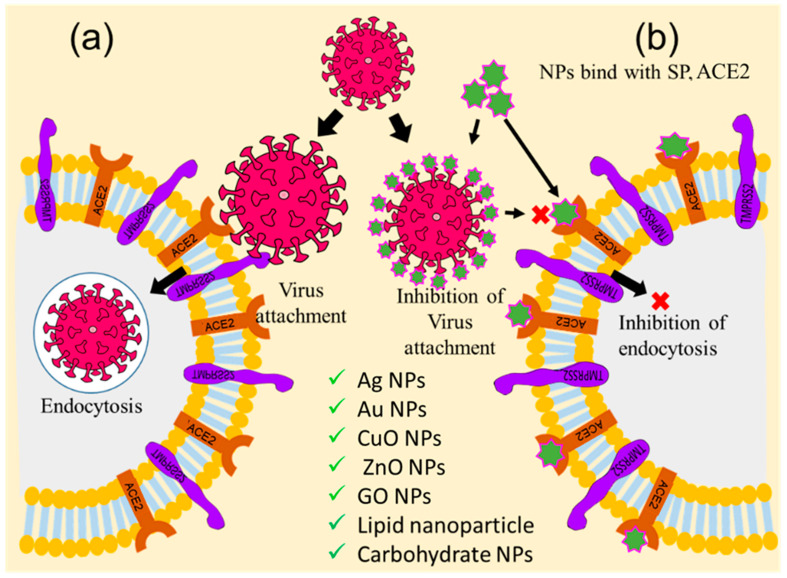
Graphical illustration of the inhibition of receptor-mediated host–virus attachments. (**a**) Receptor-mediated host–virus attachment and internalization, and (**b**) nanoparticle-mediated blocking of ACE2 for inhibition of virus attachments (‘×’ indicates inhibition checkpoint for virus attachment).

**Figure 5 ijms-24-13130-f005:**
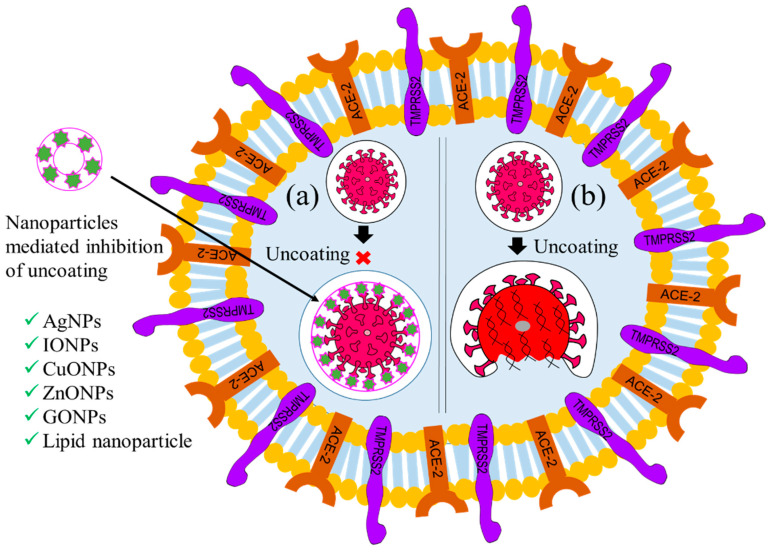
Graphical illustration of the inhibition of the uncoating of a virus. (**a**) Nanoparticle-mediated inhibition of uncoating and (**b**) uncoating in absence of NPs. (‘×’ indicates inhibition checkpoint for virus uncoating).

**Figure 6 ijms-24-13130-f006:**
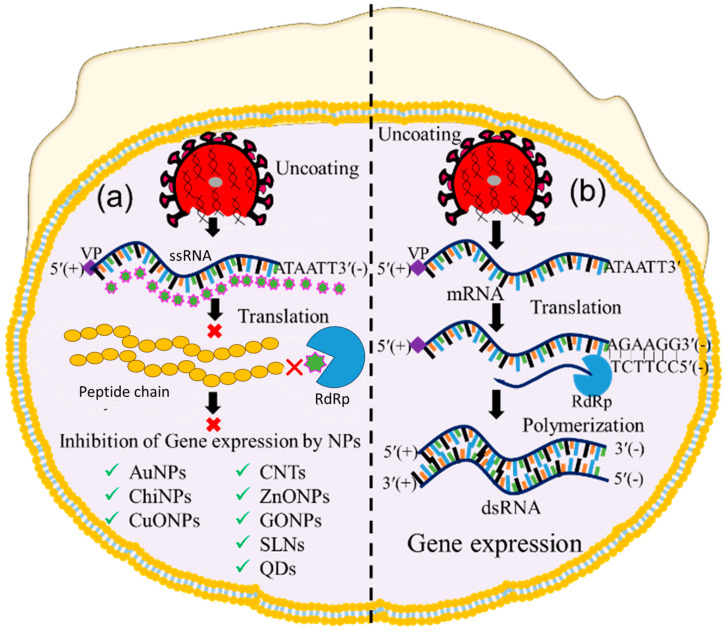
Image showing nanoparticle-assisted inhibition and no inhibition of viral gene expression. (**a**) Transcription, translation, and polymerization blocking, and (**b**) gene expression without inhibition. (‘×’ indicates inhibition checkpoint for viral gene expression).

**Figure 7 ijms-24-13130-f007:**
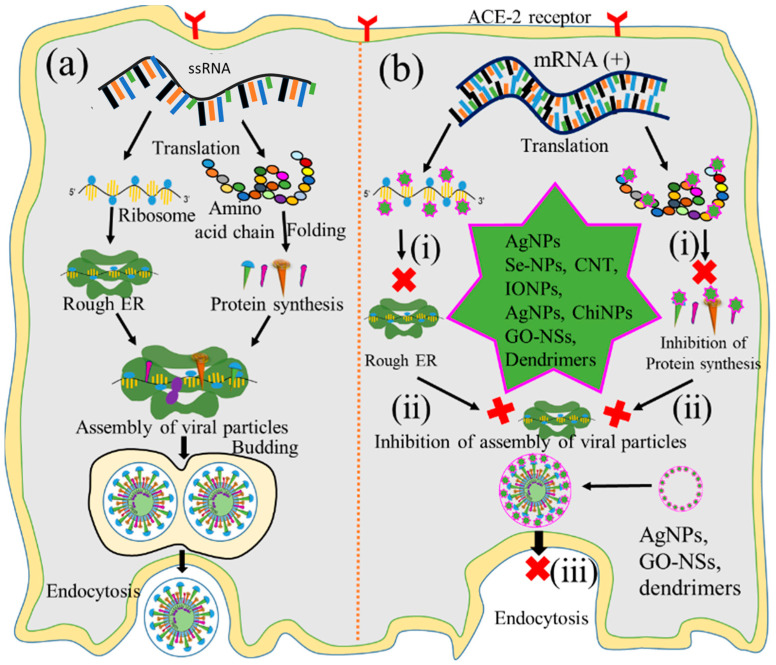
Graphical illustration of the inhibition of protein synthesis. (**a**) Protein synthesis, viral assembly, and fusion of viral replication and nanoparticle-mediated inhibition of (**b**): (**i**) protein synthesis, (**ii**) viral particle assembly, and (**iii**) release of virion. (‘×’ indicates inhibition checkpoint for proteib synthesis, Assambly and release of virus).

**Figure 8 ijms-24-13130-f008:**
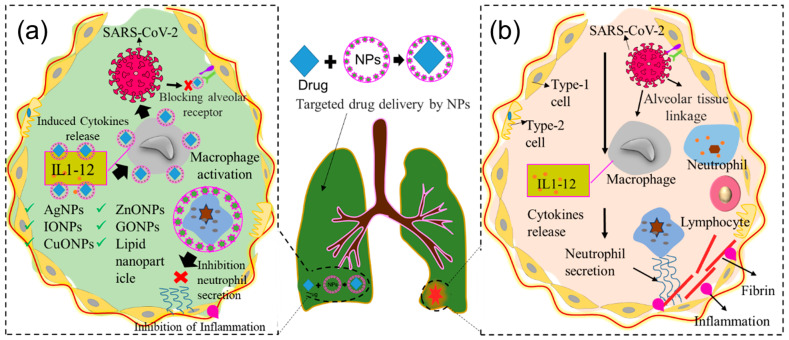
Graphical illustration of targeted drug delivery. (**a**) Viral pathogenesis at lung alveoli and (**b**) release of the drug from the lung alveoli to inhibit viral inflammation (‘×’ indicates inhibition check point).

**Figure 9 ijms-24-13130-f009:**
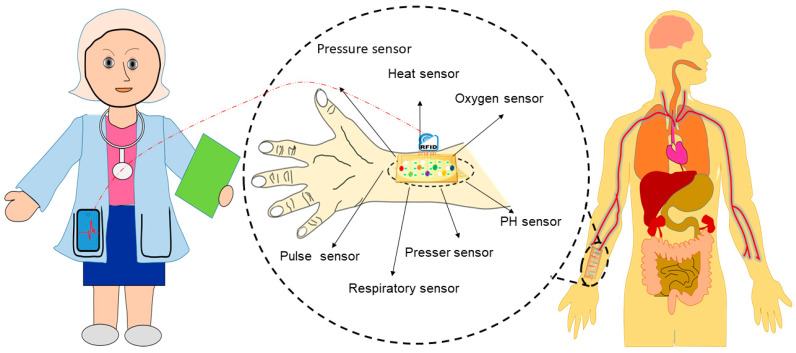
Development of wearable devices for distance monitoring of COVID-19 patients.

**Figure 10 ijms-24-13130-f010:**
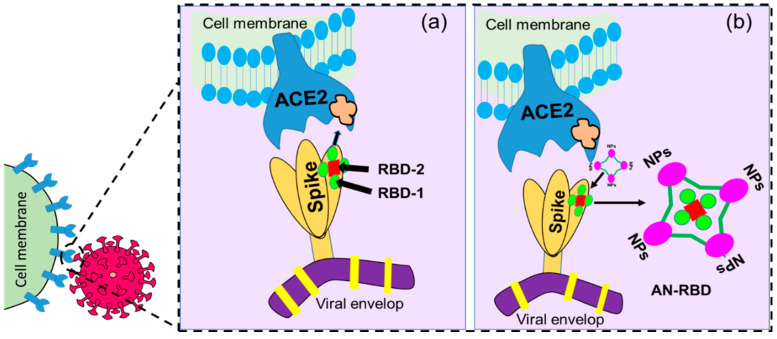
Graphical illustration of (**a**) receptor-binding domain (RBD)-mediated host cell–virus relationship where RBD-1 is the main part responsible for the binding, and (**b**) antagonistic nanostructure encapsulated RBD (AN-RBD) inhibiting adhesion.

**Figure 11 ijms-24-13130-f011:**
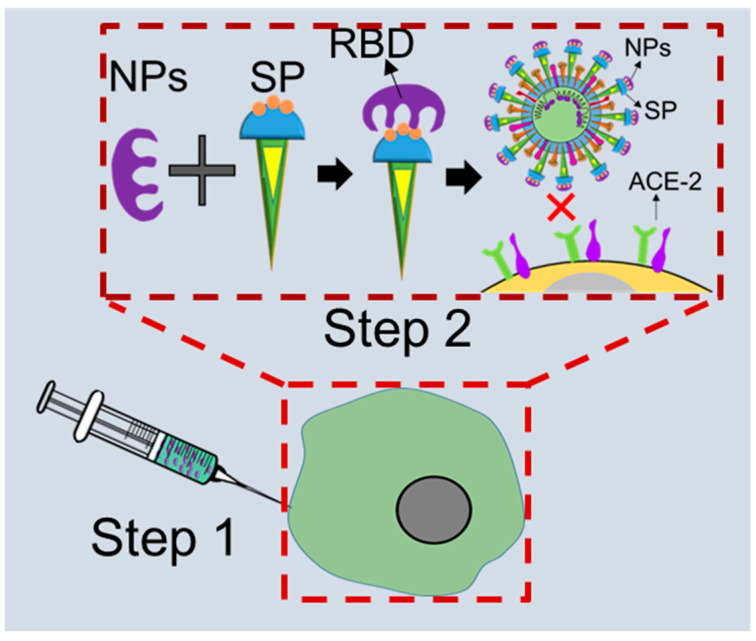
The proposed scope of COVID-19 treatment approaches: (Step 1) injection of functionalized anti-sialic receptor-like nanoparticles (NPs) and (Step 2) NPs and spike protein (SP) of SARS-CoV-2 attach and block the receptor-binding domain (RBD); SP cannot bind to angiotensin convertase enzyme (ACE2) (‘×’ indicates inhibition checkpoint).

**Figure 12 ijms-24-13130-f012:**
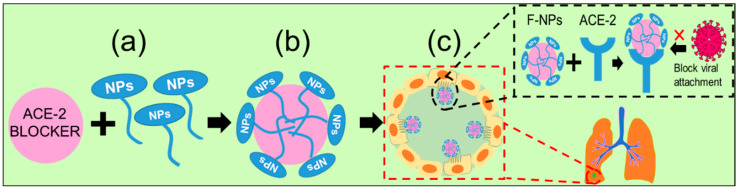
The suggested approach of targeted delivery of nanoparticles: (**a**) functionalized NPs will be conjugated to the drug or vaccine carrier, (**b**) drug-capped NPs will be carried to the targeted cells, and (**c**) ACE2 receptor of the most vulnerable lung cell will be blocked by the drug-capped NPs, which represents the nano-preventive approach for neutralizing virus in the viremia stage (‘×’ indicates inhibition checkpoint).

**Table 1 ijms-24-13130-t001:** Mutations at the RBD domain of the spike protein and their impact on immunization.

Regional Variant	Scientific Name	WHO Name	RBD Mutation							Spreading Nature	Reduced Neutralizing Ability
			Mutation Residues								
			K 417	L 452	K 477	T 478	E 484	Q 498	N 501		
United Kingdom, September 2020	B.1.1.7	Alpha	K	L	K	T	E	Q	Y	+	3-fold (Pfizer/BioNTech) and 6-fold (AstraZeneca)
South Africa, October 2020	B.1.351	Beta	N	L	K	T	K	Q	Y	++	≤86-fold (AstraZeneca) ≤6.5, ≤8.6, and ≤1.6-fold (Moderna, Pfizer-BioNTech, and Sinopharm)
Brazil and Japan, December 2020	P.1	Gamma	T	L	K	T	K	Q	Y	++	6.7- and 4.5-fold (Pfizer-BioNTech and Moderna)
India, December 2020	B.1.617.2	Delta	N	R	K	K	Q	Q	N	+	2.5-fold (Pfizer/BioNTech, Moderna, and Janssen vaccine)
South Africa, November 2021	B.1.1.529	Omicron	N	L	N	K	A	R	Y	+++	41-fold (Pfizer-BioNTech)

K = lysine, L = leucine, T = threonine, E = glutamic acid, N = asparagine, Y = tyrosine, R = arginine, Q = glutamine, and A = alanine (Red color indicates mutation sites of various amino acid residues at RBD motif for different variants). + = 50–60%, ++ = 60–80%, and +++ ≤ 80% [37,38].

**Table 2 ijms-24-13130-t002:** Different nanoparticle-based delivery systems.

Nanoparticles	Targeted Site	Action	Reference
Nanoemulsion	Monocytes, lung cells, cancer cells	Prevention from hydrolysis and oxidation allows for durable action Minimize vascular inflammation	[128]
Nanogel	Blood cell, THP-1, and HaCaT cell lines	Stability in blood circulation Enhanced anti-inflammatory action through inhibition of LOX and COX activities in cells	[129]
Nano-capsule (resveratrol-charged lipid-core-nano-capsule)	Cancer cell, HT29 cell lines	Controlled drug release Destruction of colon cancer cells Enhanced anticancer activity in HT29 cancer cells	[130]
Nanosponges (cyclodextrin drug-coated nanosponges)	Tumor cells	Increased anti-tumor activities	[131]
Chitosan	Buccal, intestinal, nasal, ocular, and pulmonary cells	Interaction with the ocular mucosa and prolonged release of the antibiotic Enhanced half-life of the drug in the eyes	[132]
Alginate	Sublingual cells	Dip in serum glucose levels and increase in serum insulin levels in diabetic rats	[133]
Xanthan gum	Buccal cells	Increased adhesion to buccal cells and release of tannin in buccal mucosa to treat diarrhea	[134]
Cellulose	Colon, nasal mucosa	Sustained release of cellulose nanocrystals Calcium alginate beads with carboxymethylcellulose (CMC)-loaded 5-fuoroacyl (5-FU); 90% release of 5-FU encapsulated in the beads Increased permeation of acyclovir into the nasal mucosa	[135]
Liposomes	cell membranes	Increased opsonization and immunogenicity of RES (reticuloendothelial system) Boost in drug delivery efficiency of the liposomes	[136]
Polymeric micelles	ARPE-19 cells, Eye tissues	Enhancement of cell proliferation, attachment, and relocation Inhibition of rear eye tissue damage	[137]
Dendrimers	Cancer cells and MCF-7 cells	Folate-attached poly-l-lysine dendrimers control cancer cell Increased concentration of doxorubicin in the tumor Increased cell uptake and low cytotoxicity in MCF-7 cell lines	[138]
Inorganic (silver, gold, iron oxide, and silica) nanoparticles	Bacteria and virus cell membranes	Control of release through biological stimuli or light activation	[139]
Nanocrystals	Pulmonary tissues	Enhanced dissolution velocity and increased glueyness to surface/cell membranes Continuous release of nanoparticles helps with swelling and shows muco-adhesive potential Enhanced inhalation efficacy under disease conditions	[140]
Quantum dots	Bone marrow cells, liver, cancer, and tumor cells	Diffusion into the entire bone marrow and labeling of rare populations of cells, such as hematopoietic and progenitor cells Attachment of an anti-GPC3-antibody to the nanoplatform results in selective separation of HepG2 hepatocellular carcinoma cells from infected blood samples	[141]

## Data Availability

Data sharing not applicable.

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
