# Peer review of "Nano-Enabled Antivirals for Overcoming Antibody Escaped Mutations Based SARS-CoV-2 Waves"

_ijms, 2023, doi:10.3390/ijms241713130_

Round 1
Author Response
첨부 파일을 참조하십시오.
우리는 모든 제안과 건설적인 의견을 고려하여 원고를 수정했습니다. 우리는 이러한 의견과 제안을 처리함으로써 우리 논문의 품질이 훨씬 향상되었다고 믿습니다.

Reviewer 2 Report
This paper extensively reviewed current applications of nanoparticles in fighting conoraviruses and other viral infections, and provided prospectives of future NP-based therapies for SARS-CoV-2 mutants.
The major concerns that I run into are:
1. In terms of SARS-CoV-2 mutant part, the review is too out of date. No information about current mutants, such as omicron BA.5/ XBB /XBB descendents were mentioned.
2. Missing an important part for a general discussion about NPs. It is advised that you include a section for NP structure, components and classifications, and add one figure of that.
3. In terms of how NPs inhibit virus, it is under-discussed. Please use more specific details / examples to discuss how NPs target each step of viral cycle. For the figure 3-6, more details are needed as well.
4. Please pay attention to the organization of your paper. Your theme is how NPs were used for fighting SARS-CoV-2 mutants, so do not include any irrelevant information on cancer, and focus on the SARS-CoV-2 more. You may want to mention conserved targets of coronavirus, such as stem-helix region and fusion peptide of Spike, and conserved epitopes of RBD.
5. Please discuss more about the advantages of NP in antiviral purposes, especially comparing with neutralizing antibodies/existing antiviral drugs.
Other minor concerns, see attached file.

Please be concise in wording. Please pay attention to spelling and formatting, especially the use of /space/, /hyphen/ /dash/ and /capital/. Check with ChatGTP or other AI tools for better language.
Author Response
Please see the attachment.
We revised the manuscript considering all suggestions and constructive comments. We believe the quality of our paper has improved much through addressing those comments and suggestions.

Round 2
Reviewer 1 Report
This revision is a great improvement on the original and clarifies the points leading to misunderstandings arising from the first version. The revisions do need some minor editing of the quality of English syntax.
The revisions need checking over by all authors. Most revisions do not need any changes but the new text is Section 4 does need some corrections.